# Ionizing Radiation Induces Resistant Glioblastoma Stem-Like Cells by Promoting Autophagy via the Wnt/β-Catenin Pathway

**DOI:** 10.3390/life11050451

**Published:** 2021-05-18

**Authors:** Cheng-Yu Tsai, Huey-Jiun Ko, Chi-Ying F. Huang, Ching-Yi Lin, Shean-Jaw Chiou, Yu-Feng Su, Ann-Shung Lieu, Joon-Khim Loh, Aij-Lie Kwan, Tsung-Hsien Chuang, Yi-Ren Hong

**Affiliations:** 1Ph.D. Program in Environmental and Occupational Medicine, College of Medicine, Kaohsiung Medical University and National Health Research Institutes, Kaohsiung 807, Taiwan; 1030459@kmuh.org.tw; 2Department of Neurosurgery, Kaohsiung Medical University Hospital, Kaohsiung 807, Taiwan; 4a1h0010@gmail.com (C.-Y.L.); 870082@kmuh.org.tw (Y.-F.S.); 770223@kmuh.org.tw (A.-S.L.); jokhlo@kmu.edu.tw (J.-K.L.); 3Graduate Institute of Medicine, College of Medicine, Kaohsiung Medical University, Kaohsiung 807, Taiwan; o870391@yahoo.com.tw; 4Department of Biochemistry, College of Medicine, Kaohsiung Medical University, Kaohsiung 807, Taiwan; sheanjaw@kmu.edu.tw; 5Department of Biotechnology and Laboratory Science in Medicine, Institute of Biopharmaceutical Sciences, National Yang-Ming University, Taipei 112, Taiwan; cyhuang5@ym.edu.tw; 6Immunology Research Center, National Health Research Institutes, Miaoli 350, Taiwan; 7Department of Biological Sciences, National Sun Yat-Sen University, Kaohsiung 804, Taiwan; 8Department of Medical Research, Kaohsiung Medical University Hospital, Kaohsiung 807, Taiwan

**Keywords:** GBM, CSC, ionizing radiation (IR), GSC, Wnt/β-Catenin, autophagy, radiation resistance

## Abstract

Therapeutic resistance in recurrent glioblastoma multiforme (GBM) after concurrent chemoradiotherapy (CCRT) is a challenging issue. Although standard fractionated radiation is essential to treat GBM, it has led to local recurrence along with therapy-resistant cells in the ionizing radiation (IR) field. Lines of evidence showed cancer stem cells (CSCs) play a vital role in therapy resistance in many cancer types, including GBM. However, the molecular mechanism is poorly understood. Here, we proposed that autophagy could be involved in GSC induction for radioresistance. In a clinical setting, patients who received radiation/chemotherapy had higher LC3II expression and showed poor overall survival compared with those with low LC3 II. In a cell model, U87MG and GBM8401 expressed high level of stemness markers CD133, CD44, Nestin, and autophagy marker P62/LC3II after receiving standard fractionated IR. Furthermore, Wnt/β-catenin proved to be a potential pathway and related to P62 by using proteasome inhibitor (MG132). Moreover, pharmacological inhibition of autophagy with BAF and CQ inhibit GSC cell growth by impairing autophagy flux as demonstrated by decrease Nestin, CD133, and SOX-2 levels. In conclusion, we demonstrated that fractionated IR could induce GSCs with the stemness phenotype by P62-mediated autophagy through the Wnt/β-catenin for radioresistance. This study offers a new therapeutic strategy for targeting GBM in the future.

## 1. Introduction

Glioblastoma multiforme (GBM) is one of the most aggressive and recurrent malignant tumors classified as Grade IV astrocytoma by the World Health Organization [1]. Despite surgery and standard concurrent chemoradiotherapy (CCRT), a poor prognosis with a mean survival duration of <15 months under recurrence indicates that GBM is therapeutically resistant [2,3,4]. Recent clinical findings showed that therapeutic resistance is associated with chemo/radio-resistant cells, which present under the recurrence condition in the ionizing radiation (IR) field [5]. Although standard fractionated radiation is effective and essential for GBM treatment, many studies have revealed that IR tends to induce a cancer stem cell (CSC)-like phenotype in other cancers, such as breast, lung, and prostate cancers, leading to therapeutic resistance [6,7,8,9,10,11].

Accumulating evidence revealed that CSCs with the self-renewal and pluripotent ability play a vital role in therapeutic resistance in different cancer types, including the GBM subpopulation [6,7,12]. These GBM-CSCs (GSCs) are characterized by the self-renewal ability in vitro (neurosphere formation) and in vivo through high expressions of neuron stem markers, such as CD133, CD44, and Nestin, as well as transcript factors, such as SOX2 and OCT4 [13,14,15]. However, the underlying molecular mechanism through which fractionated IR-induced GSCs lead to radioresistance remains unclear.

Autophagy is a cellular degradation process by lysosomes for cell renewal in a stressful environment [16]. The process is controlled by a highly conserved autophagy-related protein (Atgs)/P62/LC3II. In cancer cells, autophagy is a double-edged sword. In early stages, it could limit tumorigenesis. However, it could also provide a prosurvival function for adaptation and detoxification in a stressful environment, such as starvation, hypoxia, and chemotherapy/radiotherapy [17]. Reports by investigators have showed that autophagy activity increased after IR and chemotherapy. It is an escape mechanism for cell survival in response to cytotoxic agents, including IR and Temozolomide (TMZ) in GBM treatment [18,19]. Although recent studies have revealed that autophagy could modulate CSCs in cancers, it is unclear whether autophagy activation is involved in IR-induced/enhanced/selected GSCs to result in radioresistance.

In our previous study, we have demonstrated that P62-mediated autophagy is associated with the Wnt/β-catenin pathway in glioma cells [20]. P62 is activated during hypoxia with tumor outgrowth through β-catenin signaling inhibition. An increase in P62 protein following β-catenin knockdown and β-catenin-mediated repression of the P62 transcript highlight P62 regulation as a Wnt/β-catenin-mediated autophagy regulation mechanism [21,22]. In addition, CD133 has been found to contribute to cell survival through regulating autophagy to enhance resistance in glioma cells [23], while CD44 may be involved in positive regulation in the Wnt signaling pathway for tumorigenesis [24,25]. All these findings implicate that the Wnt/β-catenin pathway is closely correlated with GSCs and P62-mediated autophagy in GBM.

In this study, our work showed for the first time that fractionated IR could induce GSCs with the stemness phenotype in GBM. These cells participate in therapeutic resistance during tumor progression after treatment and finally contribute to recurrence and aggressive behavior with poor prognosis.

## 2. Materials and Methods

### 2.1. Cell Culture

The human GBM cell lines, U87-MG (ATCC^®^ HTB-14™; GBM of unknown origin), GBM8401 (BCRC 60163), U-251 MG (ECACC 09063001, formerly known as U-373 MG), T98G (ATCC^®^ CRL-1690™), H4 (ATCC^®^ HTB-148™), and MO59K (ATCC^®^ CRL-2365™), were obtained from the American Type Culture Collection (ATCC, Manassas, VA, USA), European Collection of Authenticated Cell Cultures (ECACC, Porton Down, Salisbury, UK; deposited by Shanghaisixin Biotech Co., Ltd., Shanghai, China) and Bioresource Collection and Research Center (BCRC, Hsinchu, Taiwan). Cells were cultured in minimum essential medium (MEM) and RPMI-1640 supplemented with 10% fetal bovine serum (FBS), 2 mM L-glutamine, and 100 U/mL penicillin and 100 mg/mL streptomycin (all from Gibco; Thermo Fisher Scientific, Inc., Waltham, MA, USA). All cells were grown at 37 °C under normoxic conditions of 95% air and 5% CO2. Cells were treated with 100 μM chloroquine (CQ, Sigma, C6628, St. Louis, MO, USA) or 100 nM bafilomycin (BAF, Sigma, B1793, MO, USA) for 24 h.

Human brain tissues: We recruited 106 patients with stage I–IV glioma in Department of Neurosurgery, Kaohsiung Medical University Hospital between 2000–2011. The protocols and ethics statements in this study were approved by the Institutional Review Board of Kaohsiung Medical University Hospital (No. KMUHIRB-20110185 and No. KMUHIRB-F(I)-20200024). Informed consent was obtained from all subjects involved in the study data on the patient’s basic profile; age, gender, therapy type, Karnofsky Performance scale, and tumor grading were all included. We also examined the LC3II score by a scoring system (from 0,1,2,3) and compared it to our clinicopathological findings. LC3II expression was graded by a scoring system (0 is the frequency of nucleus staining if <10% of the cell is positive; 1 is the frequency of nucleus staining if >10% of the cell is positive and intensity is low; 2 is the frequency of nucleus staining if >10% of the cell is positive, and intensity is moderate; 3 is the frequency of nucleus staining if >10% of the cell is positive and intensity is strong). All patients were also separated into two groups as radiation or chemotherapy (radiation/chemotherapy) and without any therapy. Temozolomide (TMZ) was used as chemotherapy. We discuss the details in the results.

### 2.2. Cell Viability Assay

U87MG and GBM8401 cells were seeded (2000 cells/well) in a 96-well plate. After 24 h of culture, the cells were exposed to radiation with doses of 2 Gy for 1 day, 2 Gy for 5 days, and 10 Gy for 1 day. After 24 h, the cell viability was measured by the CCK-8 kit (CCK-8, sigma 96992, Sigma-Aldrich; Merck KGaA, Darmstadt, Germany) at 450 nm according to the manufacturer’s instructions. This experiment was repeated at least three times.

### 2.3. Long Term Survival and Colony Formation Assay

Cells were seeded into 6-well plates (5 × 10^2^ cells/well). After 24 h of culture, the cells were exposed to radiation with doses of 2 Gy for 1 day, 2 Gy for 5 days, and 10 Gy for 1 day. After treatment, cells were cultured at 37 °C for 7–21 days. The cells were washed twice with PBS, fixed in 4% paraformaldehyde for 30 min, and stained with 0.1% crystal violet for 20 min at 25 °C. The colonies were carefully washed with tap water, and then the number of colonies, defined as >50 cells/colony, was counted and analyzed. Then the cells were washed with PBS and DMSO was added to induce a complete dissolution of the crystal violet. Absorbance was recorded at 570 nm by a 96-well-plate ELISA reader. Results were expressed as average colony count ± SE from three independent experiments.

### 2.4. Quantitative Real-Time RT-PCR (TaqMan Assay)

The ABI Prism^®^ 7700 Sequence Detection System (Applied Biosystems, Foster City, CA, USA) was used for quantitative analysis of mRNA expression. Total RNA was isolated using a Tissue Total RNA Mini Kit (Geneaid, Taipei, Taiwan) according to the manufacturer’s instructions. In-column DNase I digestion was performed to remove genomic DNA contamination. Reverse transcription was then carried out using Superscript IV (Invitrogen Life Technologies, Carlsbad, CA, USA) and Oligo(dT)20 primers following the manufacturer’s guidelines. Gene expression was quantified using Taqman Gene Expression Assay reagents (Applied Biosystems, Foster City, CA, USA) following the procedures provided by the manufacturer with 18s as the inner reference. The threshold cycle (Ct) values were calculated using the StepOnePlus system software (Applied Biosystems, Foster City, CA, USA). The relative expression of each mRNA was calculated by the 2−(ΔCt) method. The primer sequences were as follows: CD133 (Homo sapiens pan paniscus prominin 1, PROM1/CD133, NCBI reference sequence is XM_034958337.1) forward 5’-GAGCTAAGGGAAGGGCGG-3’; CD133 reverse 5’-TTCTGTCTGAGGCTGGCTTG-3’; CD44 (Homo sapiens pan paniscus CD44, NCBI reference sequence is XM_034932399.1) forward 5’-CTGCAGGTATGGGTTCATAG-3’; CD44 reverse 5’-ATATGTGTCATACTGGGAGGTG-3’; Nestin (Homo sapiens trachypithecus francoisi nestin, NES, NCBI reference sequence is XM_033196923) forward 5’-GGGAAGAGGTGATGGAACCA-3’; Nestin reverse 5’-AAGCCCTGAACCCTCTTTGC-3’; P62 (Homo sapiens sapajus apella sequestosome 1, SQSTM1/P62, NCBI reference sequence is XM_032270231.1) forward 5’-TACCAGGACAGCGAGAGGAA-3’; P62 reverse 5’-TCCTTTCTCAAGCCCCATGT -3’; LC3II (Homo sapiens microtubule associated protein 1 light chain 3 beta 2, MAP1LC3B2, NCBI reference sequence is NM_001085481) forward 5’-GATGTCCGACTTATTCGAGAGC-3’; LC3II reverse 5’-TTGAGCTGTAAGCGCCTTCTA-3’; 18s (Homo sapiens DIMT1 rRNA methyltransferase and ribosome maturation factor, DIMT1/18s, NCBI reference sequence is NM_001348076.2) forward 5’-TCAAGTGCAGTGCAACAACTC-3’; and 18s reverse 5’-AGAGGACAGGGTGGAGTAATCA-3’.

### 2.5. Immunoblot Analysis

Protein expression was detected by immunoblot analysis. Cell lysates with equal protein content were prepared in SDS sample buffer, separated on NativePAGE Novexw Bis–Tris 4–16% gel for BN-PAGE analysis (Invitrogen; Thermo Fisher Scientific, Inc.) followed by transferring to polyvinylidene fluoride membranes. Proteins on the membrane were detected with specific primary antibodies and HRP-conjugated secondary antibodies. The following antibodies were used: P62 (1:2000; ab56416, Abcam, Cambridge, MA, USA), β-actin (1:1000; ab3280, Abcam, Cambridge, MA, USA), LC3II (1:1000; AP1802a, Abgent, San Diego, CA, USA), Fzd 1 (1:1000; GTX108181, GeneTex, Irvine, CA, USA), GAPDH (1:10000; GTX100283, GeneTex, Irvine, CA, USA), GSK3β (1:1000; BD610202, BD Biosciences, San Jose, CA), GSK3β Ser9 (1:1000; #9323s, Cell Signaling Technology, Inc.), β-catenin (1:1000; #9562s, Cell Signaling Technology, Inc.), β-catenin Ser33/37/41 (1:1000; #9561s, Cell Signaling Technology, Inc.), CD133 (1:1000; #5860s, Cell Signaling Technology, Inc.), CD44 (1:1000; ab157107, Abcam, Cambridge, MA, USA), Nestin (1:1000; ABD69, Millipore Sigma, Burlington, MA), SOX-2 (1:1000; ab97959, Abcam, Cambridge, MA, USA), and β-actin (1:1000; sc47778, Santa Cruz Biotechnology, Santa Cruz, CA). The signal of each target protein was visualized by incubation with ECL Reagent and exposure to X-ray film.

### 2.6. Immunohistochemical (IHC) Staining

The IHC staining was performed on 4 μm paraffin sections. The sections were dewaxed, hydrated, and placed at 4 °C overnight. For antibodies against CD133 (AP1802a, Abgent, San Diego, CA, USA), P62 (ab56416, Abcam, Cambridge, MA, USA) and LC3II (AP1802a, Abgent, San Diego, CA, USA), the standard avidin-biotin complex (ABC) procedures were employed. After the sections were returned to room temperature, biotinylated secondary antibodies and horseradish-labeled streptavidin were added. The samples were then incubated in an oven at 37 °C Subsequently, DAB color development, hematoxylin counterstaining, gradient alcohol dehydration, and xylene transparent were carried out. All samples were sealed with neutral gum afterwards.

### 2.7. Databases

To identify significant differentially expressed genes (DEGs) after irradiation, three whole gene expression databases, which covered cell base, xenograft animal model, and clinical GBM specimens from National Center for Biotechnology Information (NCBI) Gene Expression Omnibus (GEO), were explored. Data of gene chip GSE107040, GSE117126, and GSE82139 were obtained from GEO database. GSE107040 was from Life Science in the department of Research Institute for Natural Sciences at Hanyang University with three cases of U87MG_non-irradiated as a control group and three cases of U87MG_2Gy × 1 as experimental group. GSE117126 was from Neurology in the Department of Medicine at Seoul National University with one case of brain nonirradiated as a control group and one case of brain irradiated in orthotopic U-87 MG xenograft mouse models. GSE82139 was from Glioma and Neural Stem Cell Group in the Department of Institute Catalan d’Oncologia IDIBELL at Barcelona with two cases of normal samples as a control group and two cases of GBM patients as the experimental group. To obtain the P-value, multiple testing corrections were applied with the Benjamini–Hochberg method. Only those genes exhibiting log2 fold change (FC) greater than 1.5 and adjusted *p* < 0.05 were considered to be DEGs by Ingenuity Pathway Analysis (IPA) software. The details of the significant probe sets are summarized in Table 1. The comparison resulted in a list of 17 autophagy-related DEGs. Among them, LC3 II, P62, CTSF, and VPS33A were consistently upregulated in all three databases.

### 2.8. Statistical Analysis

The Western blot protein bands were quantified via densitometry using Multi Gauge 3.0 program (Fujifilm, Tokyo, Japan). For the in-vitro and in-vivo studies, statistical significance was evaluated through one-way analysis of variance (ANOVA) followed by Tukey’s post hoc test to correct for multiple comparisons. A two-tailed Student’s *t*-test was used to compare data between two groups. Statistical significance was set as *p* < 0.05. The association between P62, LC3 II, and CD133 expressions, and clinicopathological characteristics of GBM patients were investigated using ANOVA test, Fisher’s exact test, as well as Chi-squared test. Survival analysis was performed using the Kaplan–Meier method to calculate overall and disease-free survival rates among different groups. These groups were then compared using log-rank method. Two-sided *p* < 0.05 was considered as a statistically significant difference.

## 3. Results

### 3.1. Correlation of Clinicopathological Parameters with Autophagy and GSC Markers in GBM Samples

In total, there were 56 males and 50 female patients without gender variants. Sixty-four patients were over the age of 65 years old and 42 patients were lower than 65 years old. Tumor grading is 10 patients for grade I (pilocytic astrocytoma), 16 patients for grade II (astrocytoma), 17 patients for grade III (anaplastic astrocytoma), and 63 patients for grade IV (GBM). Fifty-seven patients received radiation/chemotherapy and 49 patients did not receive any therapy. Retrospective analysis showed that GBM patients who received radiation/chemotherapy had significantly higher levels of LC3II expression, when compared with those without receiving radiation or chemotherapy (*p* = 0.021, Table 2), meaning autophagic markers are related to radiation/chemotherapy. Although tumor grading had no significant difference in statistical level, it still had an increasing tendency with LC3II scores with high tumor grade. Grade I (pilocytic astrocytoma) had a much lower expression score of LC3II, and Grade IV (GBM) predominately had a higher expression score of LC3II. The relationship between the expression level of CD133 in correlation with LC3II and the prognosis of patients with glioma was also investigated. High levels of both CD133 and LC3II expression showed significantly shortened survival times (Figure 1A).

We further examined the specimens of pre- and post-CCRT in two representative GBM patients by immunohistochemical (IHC) staining. High expression levels of CD133/CD44/ LC3II/P62, were found in post-CCRT samples, whereas low expression levels of CD133/CD44/ LC3II/P62, were shown in pre-CCRT ones (Figure 1B,C). Collectively, we found that the patients who received radiation/chemotherapy had high LC3II/P62 and CD133/CD44 expression, indicating the possible correlation of radiation/chemotherapy with autophagy as well as the GSC phenotype in GBM.

### 3.2. Pathway Analysis for Potential Involvement of Genes Associated with Autophagy and GSC Markers

To explore possible mechanisms and potential gene involvement, we used three publicly available microarray datasets from the National Center for Biotechnology Information Gene Expression Omnibus (GEO; Table 1) and identified differences in the expression of autophagy-related genes between non-irradiation and irradiation treatments from bench to bedside. Subsequently, autophagy-related gene expression profiles were screened within differentially expressed genes (DEGs) between the non-irradiation and irradiation treatment groups from in vitro and in vivo experiments to the clinic. After data processing, 17 DEGs were identified between non-irradiation and irradiation treatments from in vitro and in vivo experiments to the clinic. These DEGs contained nine upregulated and eight downregulated genes. Among them, LC3II, P62, Cathepsin F (CTSF), and VPS33A Core Subunit of CORVET and HOPS Complexes (VPS33A) were consistently upregulated in all databases (Appendix A). According to the CD133 expression level increased in two databases (Table 1), these data consist with our hypothesis that GSC stemness was associated with autophagy. Afterwards, we mapped these DEGs on the autophagy pathway using ingenuity pathway analysis (IPA, QIAGEN, CA, USA. Website: www.ingenuity.com, (accessed on 23 December 2019) software. These genes were color coded based on the cutoff of an absolute correlation coefficient.

In consideration of the correlation of CD133 and LC3II expression in clinicopathologic parameters/samples and the GEO database archives of GBM, we hypothesized that potential autophagy-related genes could play a role in radiation resistance. Therefore, we examined endogenous P62 and LC3II proteins in different GBM cell lines, including U251, T98G, U87MG, H4, GBM8401, and MO59K. Immunoblotting assay against anti-P62 and anti-LC3II antibodies showed that the expressions of P62 and LC3II proteins were lower in U87MG and H4 cell lines than in other GBM cell lines (U251, T98G, GBM8401, and MO59K; Appendix A). Thus, all subsequent experiments were performed using GBM8401 and U87MG cells lines (Appendix A).

Clinical reports showed that the effects of IR were dependent on the total dose and fractionation ratio [26,27]. In the present study, we followed published clinical guidelines and used 2 Gy per day for 5 days per week [3]. Therefore, we designed our radiation experiments in dose- and fraction-dependent manners with 2 Gy (2 Gy for 1 day), 4 Gy (2 Gy per day for 2 days), 6 Gy (2 Gy per day for 3 days), 8 Gy (2 Gy per day for 4 days), 10 Gy (2 Gy per day for 5 days), and 10 Gy alone for 1 day (Appendix A). Cell viability results showed that IR affected cell growth in dose- and fraction-dependent manners at 24, 48, and 72 h, ensuring a reliable model for IR effects. From these preliminary results, IR models 2 Gy × 1, 2 Gy × 5 times, and 10 Gy alone in GBM8401 and U87MG were chosen in order to perform subsequent experiments combining dose- and fraction-dependent manners (Figure 2A). The clonogenic assay was performed to verify the cytotoxic effects of IR on GBM8401 and U87MG after IR at 7, 14, and 21 days; the results revealed that GBM cells after IR regrowth appeared as non-IR treated cells (Figure 2B). The results suggest that GBM cancer cells were recurrent and radioresistant, implying the possibility of inductive IR-resistant glioma cells.

### 3.3. Co-Expression of Autophagy and GSC Markers in GBM8401 and U87MG Cells

To investigate whether inductive IR-resistant glioma cells in our experiments were related to autophagy and GSCs, we examined the expressions of P62 and LC3II as autophagy markers and CD133, CD44, and Nestin as GSC markers in two inductive IR-resistant glioma cell lines (GBM8401 and U87MG) irradiated with 2 Gy × 1 time dose and 2 Gy × 5 times dose. The results showed that mRNA levels of autophagic markers (P62/LC3II) and GSC markers (CD133/CD44/Nestin) increased in a dose-dependent manner, indicating that autophagy and stemness are correlated with the IR accumulative dose (Figure 3A). In addition, these autophagic markers (P62/LC3II) and GSC markers (CD133/CD44/Nestin) were also evaluated in a fraction-dependent manner with 2 Gy × 5 times (10 Gy total) and 10 Gy single dose (Figure 3B,C). The data revealed that autophagy was predominant in the 2Gy × 5 group, where survival and regrow of cells were observed. In contrast, cell death without regrowth in the single dose 10 Gy group was seen under the same conditions. Taken together, although all these markers (GSC markers and autophagic markers) were induced in GBM8401 and U87MG, there are not obviously different expression levels between these two cell lines.

Furthermore, Western blot analysis for P62/LC3II, CD133/CD44/Nestin and SRY (sex determining region Y)-box2 (SOX2; GSC transcript factor) were performed under both dose- and fraction-dependent manners in irradiated U87MG and GBM8401 cells (Figure 3D). The results revealed that autophagy and GSC markers increased in a fraction-dependent manner rather than a dose-dependent manner. The results also showed that GBM8401 cells had higher expression of the autophagic markers (P62/LC3II) and GSC markers (CD133/CD44/Nestin/SOX2) than did U87MG cells.

### 3.4. Expression of the Wnt/β-Catenin/GSK3β Pathway in IR-Induced GSC-Like Phenotype Cells

The aforementioned results demonstrated that IR could induce GSCs with a stemness phenotype through autophagy in both dose- and fraction-dependent manners in two glioma cell lines. However, the involved pathway was still unclear. As presented in Figure 4A, IR reduced the protein levels of frizzled (indirectly), β-catenin (directly), and phospho-β-catenin (Ser33/Ser37/Thr41) through phosphorylation by GSK3β in GBM8401 and U87MG cells in dose- and, more noticeably, fraction-dependent manners. Moreover, the phosphorylation level of GSK3β (S9) decreased with IR, indicating that GSK3β activity increased and phosphorylated β-catenin.

To further verify whether β-catenin protein decreased due to protein degradation, the proteasome inhibitor MG132 was applied to IR-treated cells. As expected, MG132 application reversed IR-induced β-catenin degradation and slightly suppressed P62 expression (Figure 4B). These results clearly demonstrated that IR enhanced GSK3β activity through Ser9 phosphorylation downregulation, which in turn enhanced β-catenin phosphorylation at Ser33/Ser37/Thr41, triggering protein degradation. We concluded that IR induced GSCs with the stemness phenotype through autophagy with P62-mediated β-catenin degradation through the Wnt/β-catenin signaling pathway in glioma cell lines, especially in GBM8401 cells, in a fraction-dependent manner.

### 3.5. Modulation of Autophagy in IR-Induced GSC-Like Phenotype Cells In Vitro: Effect of CQ and BAF

To confirm that autophagy was involved in IR-induced GSCs with the stemness phenotype, BAF and CQ were used as autophagy inhibitors to determine whether or not autophagy was involved in the pathway. The result show decreased viability of U87MG and GBM8401 cells in the presence of BAF or CQ, compared with radiation only (Figure 5A). The effect of irradiation on colony formation in U87MG and GBM8401 cells with or without 72 h BAF or CQ treatment was assessed by the clonogenic assay (Figure 5B). BAF or CQ treatment clearly reduced colony formation for both cell types. Finally, transcript levels of autophagy and GSCs-related genes were determined by Western blotting in irradiated cells with or without BAF or CQ treatment. The results revealed that the expression levels of LC3II and P62 were reduced in irradiated GBM8401 cells with BAF or CQ treatment in IR dose- and fraction-dependent manners, when compared with GBM8401 cells treated with IR only. Similarly, decreased expression levels of GSC markers, CD133, Nestin, and SOX2, except for CD44, in irradiated GBM8401 cells incubated with BAF or CQ were also detected, comparing with the same cells treated with IR only (Figure 5C). In all experiments, the correlation between autophagy and GSC markers was stronger in GBM8401 than in U87MG.

## 4. Discussion

GBM is one of the most aggressive tumors with poor prognosis. For more than a decade, standard clinical therapy has followed the published guidelines, which have combined TMZ and radiation (total 60 Gy in 30 daily fractions of 2 Gy), and the results have shown improved survival times from 12.1 to 14.6 months [3,4]. However, poor prognosis has persisted due to an inevitable high recurrence rate despite complete CCRT treatment. The recurrence condition was regarded as therapeutic resistance. Evidence showed that chemo- or radio-resistant cells provided therapeutic resistance and existed in the IR-field area [5]. It has been reported that the canonical Wnt signaling (also known as Wnt/β-catenin) pathway plays an essential role in stem cell fate decision. Frizzled is the upstream protein of the Wnt signaling pathway. It also regulates the relative stability of β-catenin through GSK3β-dependent phosphorylation. P62-mediated autophagy was found to be enhanced through not only the AMPK activity but also the Wnt/β-catenin signaling pathway to regulate glioma cells [20,28,29]. Therefore, we postulated that the Wnt/β-catenin signaling pathway could regulate P62-mediated autophagy and used phospho-β-catenin (Ser33/Ser37/Thr41) as β-catenin active forms for detecting the β-catenin status. GSK3β (S9) was used for β-catenin phosphorylation to degrade β-catenin. In addition, increasing reports have demonstrated that IR could induce GSCs in different tumor types [30]. Thus, we hypothesized that standard fractionated IR (fraction-dependent manner) could induce GSCs for therapy resistance. Herein, we reported for the first time that fractionated IR (fraction-dependent manner as to the clinical daily used dose: 2 Gy per day) induced GSCs with the stemness phenotype by p62-mediated autophagy through the Wnt/β-catenin/GSk3β/P62 axis signaling pathway in human glioma cells.

CSCs, also known as tumor-initiating cells or tumor-propagating cells, are a small subpopulation of cancer cells with the capacity for self-renewal and pluripotency [31]. These capacities led to differentiation and tumor heterogeneity for resisting therapy. CSCs were first identified in acute myeloid leukemia [32,33], and the first solid tumor CSC was found in glioma tumor [13,14,34]. Since then, CSCs have been identified in many cancers, such as breast [35], pancreatic [36,37], colon [38,39], and lung [40,41]. Two main CSC models have been proposed to explain the origin of CSCs: stochastic and hierarchical models. The stochastic model is based on the concept that all tumor cells are capable of producing new cancer cells through converting the non-CSC phenotype to the CSC phenotype under specific conditions [42,43]. By contrast, the hierarchical model considers that a small unique subpopulation of tumor cells known as CSCs give rise to tumor cells without the conversion concept (Figure 6A) [32]. However, the issue of which of the two models to apply has been debated for decades. Studies have identified CD44^+^CD24^−^ in breast cancer [44]; CD44^+^CD24^+^EpCAM^+^ in pancreatic or ovarian cancer [45,46]; and a high expression of neuron stem markers, such as CD133, CD44, and Nestin, and transcript factors, such as SOX2 and OCT4, in GBM. In the present study, we used CD133/CD44/Nestin/SOX2 as GSC markers and attempted to distinguish the two CSC models. Our results showed that GSC markers increased in GBM tissues, and they were highly correlated to GBM after CCRT in IHC specimens. Furthermore, our experiments showed increased protein expression of GSC markers in both IR dose- and fraction-dependent manners in the two GBM cell lines. Although our data revealed that CD133/Cd44/Nestin/SOX2 was increased by IR, whether a stochastic model or hierarchical model is more suitable for GSC by IR still needs evidence to prove.

Autophagy is considered a double-edged sword due to its dual functions in tumors. It suppresses tumor growth in the initial stage and promotes growth or survival in a stressful environment. Increasing evidence has revealed that autophagy is associated with therapeutic resistance in a stressful environment. In addition, studies have shown that autophagy in CSCs mediates therapeutic resistance [17,18,19]. Our study revealed that autophagy increased with GSC markers, including CD133, CD44, Nestin, and SOX2 and correlated with IR in dose- and fraction-dependent manners, especially the fraction-dependent manner (clinical manner). It has been reported that multiple sequential steps are involved in autophagy influx, such as sequestration, transport to lysosomes, degradation, and utilization of degradation products. Therefore, different autophagy inhibitors could inhibit in different steps and present different results in the target proteins. In our study, two different inhibitors (BAF and CQ) were used as autophagy inhibitors to determine the participation of autophagy. The data showed that the levels of GSC markers, including CD133, CD44, Nestin, and SOX2, were reduced in the presence of autophagy inhibitors. The results support that autophagy and GSCs with the stemness phenotype are induced in response to cytotoxic agents. In addition, from cell viability studies, the GBM8401 cells could survive through autophagy and present as inductive IR-resistant glioma cells with the GSC-like phenotype. Furthermore, in the colony assay, it was found that inductive IR-resistant glioma cells could present more regrowth and exhibit recurrence and radioresistance in GBM8401 cells than in U87MG cells. Overall, we believe that the inductive IR-resistant glioma cells could be regarded as IR-induced GSCs with the stemness phenotype through autophagy, especially GBM8401 cells, in a fraction-dependent manner.

Similar to our previous reports, our study found that the Wnt/β-catenin/P62 axis is a potential signaling pathway for glioma cells in therapy selection [20]. Therefore, we used this signaling pathway to elucidate the possible signaling pathway for IR-induced GCSs. We examined β-catenin, phospho-β-catenin (Ser33/Ser37/Thr41), GSK3β, and GSK3β (S9). Furthermore, we used MG132 as a proteasome inhibitor. The results demonstrated that IR enhanced GSK3β activity through Ser9 phosphorylation downregulation, which in turn enhanced β-catenin phosphorylation at Ser33/Ser37/Thr41, triggering protein degradation. Moreover, other studies have reported that β-catenin could translocate to the cell membrane for stabilizing CD133, and CD44 could regulate Wnt activity [23,24,25]. In addition, CD133 could be recycled through the autophagy process [47]. These concepts could offer an explanation regarding the crosstalk between CD133 or other GSCs markers and Wnt/β-catenin/P62. Overall, we attempted to elucidate the IR-induced GCS stemness phenotype through autophagy with P62-mediated β-catenin degradation through the Wnt/β-catenin signaling pathway in glioma cell lines, especially in a fraction-dependent manner.

On the basis of our observations, we suggest a working model for frizzled/β-catenin/P62/GSCs markers (Figure 6B). Our results implicated that the stochastic model may be more suitable for determining the origin of GSCs due to different inducible levels of GSCs and increasing the expression of GSC markers to present GBM heterogeneity. Under normal conditions, cancer cells turn on the canonical Wnt/β-catenin pathway to disintegrate the destruction complex (Axin, APC, GSK3β, and β-catenin; dash circle) and stabilize β-catenin. Subsequently, β-catenin binding to TCF in the nucleus upregulates target genes to present proliferation capacity and suppress P62 activation. However, under stress conditions (such as IR, hypoxia, or chemotherapy), cancer cells survive through shutting down the Wnt/β-catenin pathway to degrade β-catenin and then induce P62-mediated autophagy. Therefore, GBM cancer cells could shift cancer cells from the proliferation state to hibernation state through autophagy as a survival function, followed by enhancement of CD133/CD44/Nestin/SOX2 as GSC markers to support the stochastic model. Moreover, our data revealed different expression levels in the two glioma cell lines. GBM8401 cells are more predominant than U87MG cells in inducing/enhancing GSCs with the stemness phenotype. Thus, P62-mediated autophagy via Wnt/β-catenin/GSk3β/P62 axis may play a vital role for IR-induced GSCs with the stemness phenotype, especially GBM8401. It has been previously reported that GBM8401 is a P53-mutant type and U87 is a P53-wide type [20]. We speculate that the P53 status for radiosensitivity may have a role in IR-induced GSCs with the stemness phenotype for further study in the future.

## 5. Conclusions

The fractionated IR could induce the stemness phenotype in GSCs with P62-mediated autophagy through the Wnt/β-catenin signaling pathway. IR-induced GSCs provide therapeutic resistance for tumor progression after treatment and thus contribute to recurrence and aggressive behavior. Our results could aid in developing a new therapeutic strategy for GBM treatment in the future.

## Figures and Tables

**Figure 1 life-11-00451-f001:**
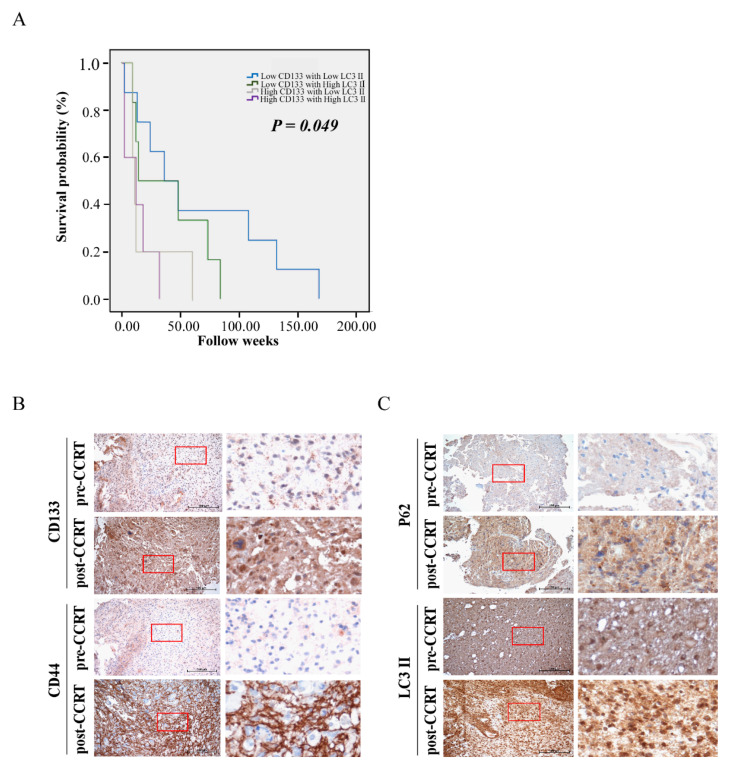
Correlation of clinic-pathological parameters with autophagic markers and CSCs markers in GBM samples. (**A**) Kaplan–Meier curves displaying the estimated survival probability for four groups (low-CD133/low-LC3II; low-CD133/high-LC3II; high-CD133/low-LC3II; high-CD133/high-LC3II) in GBM. Each vertical step in the curve indicates one or more events (i.e., deaths), and right-censored patients are indicated by a vertical mark in the curve at the censoring time. (**B**) Immunohistochemical staining for CSCs-related genes (CD133 and CD44) in two representative human primary GBM (pre-CCRT) and recurrent GBM tumor tissues (post-CCRT) surgical biopsies. (**C**) Immunohistochemical staining for autophagic genes (P62 and LC3II) in two representative human primary GBM (pre-CCRT) and recurrent GBM tumor tissues (post-CCRT) surgical biopsies. Images on the right represent magnifications of the notified areas in the corresponding left images.

**Figure 2 life-11-00451-f002:**
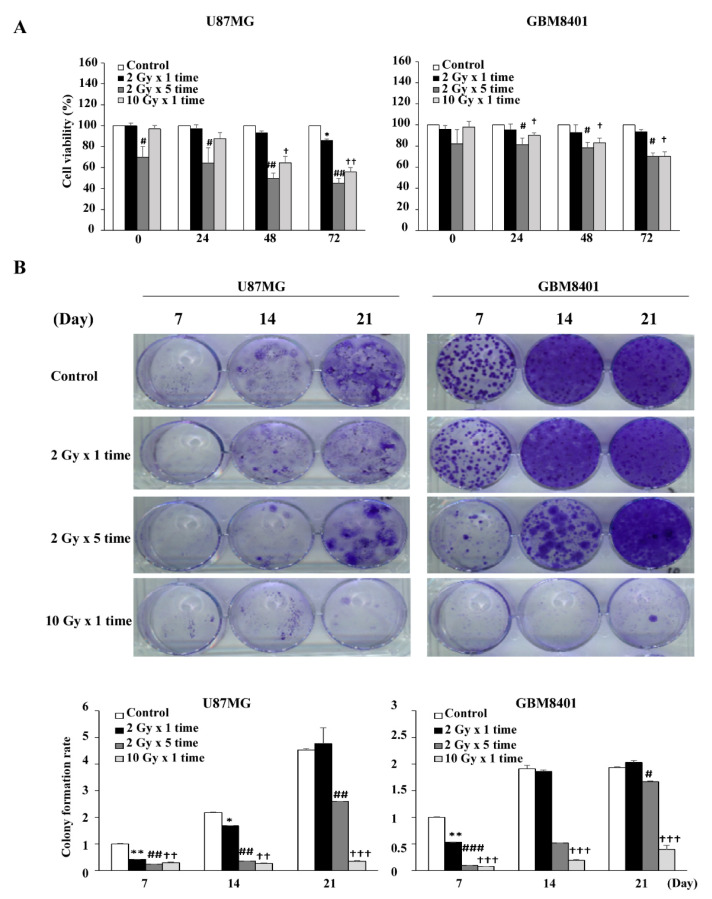
Establishing a radiation-resistant model using U87MG and GBM 8401 cells. (**A**) Irradiation suppressed the proliferation of glioma cell lines. Cells were incubated for 24, 48, and 72 h in the presence of irradiated cells (not clear), after which cell viability was assessed using MTT assays. U87MG and GBM8401 cells were divided into the following four groups: control (0 Gy), 2 Gy × 1 times (2 Gy-fraction), 2 Gy × 5 times (10 Gy-fraction), and 10 Gy × 1 times (10 Gy-only). (**B**) Clonogenic assays were performed to assess the effect of irradiated cells on colony formation. Image showing colonies produced by U87MG and GBM8401 cells following plating of 500 cells and 7–21 days incubation. Cell numbers were quantified and the error bar indicates mean ± SEM of three independent experiments. The level of significance was determined using Student’s *t*-Test with ns representing *p* > 0.05, ** *p* < 0.01, and * *p* < 0.05 compared with the 2 Gy × 1 times (2 Gy-fraction) group. # *p* < 0.05, ## *p* < 0.01, and ### *p* < 0.005 compared with the 2 Gy × 5 times (10 Gy-fraction) group. † *p* < 0.05, †† *p* < 0.01, and ††† *p* < 0.005 compared with the 10 Gy × 1 times (10 Gy-only) group.

**Figure 3 life-11-00451-f003:**
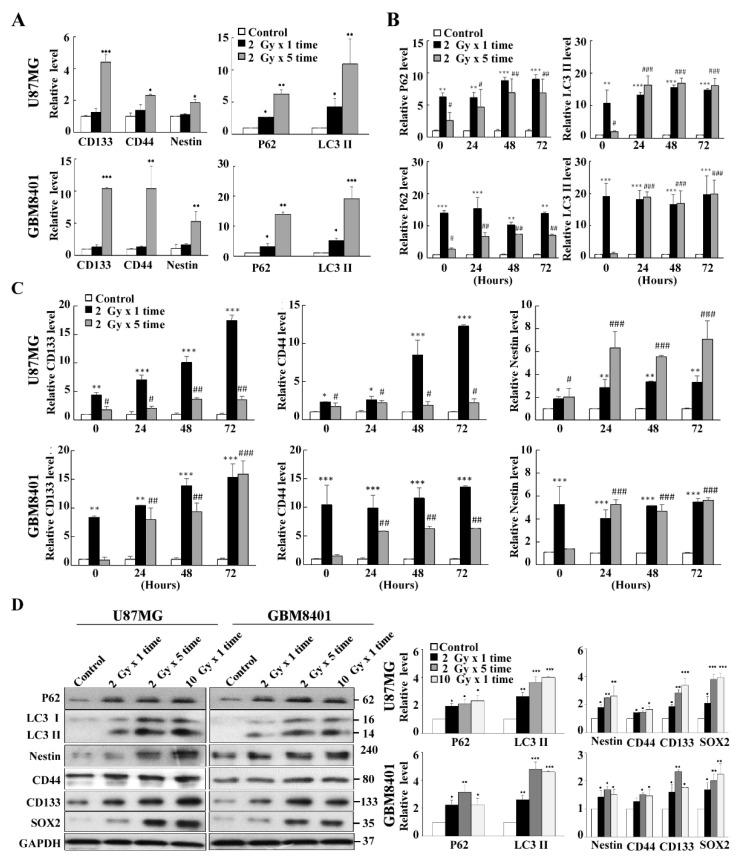
Autophagy-related candidate genes selection and validation from radiation-resistant U87MG and GBM 8401. (**A**) Quantification of CSCs-related genes (CD133, CD44 and Nestin) and autophagic genes (P62 and LC3II) after receiving radiation on U87MG and GBM 8401 cell line. All cells were treated with either a single dose of 2Gy or five fractions of 2Gy, and then incubated for 72 h. (**B**) Temporal alteration of the expression of autophagic genes (P62 and LC3II) after receiving five fractions of 2Gy or a single dose of 10Gy radiation on U87MG and GBM 8401 cells. (**C**) Temporal alteration of the expression of CSCs-related genes (CD133, CD44 and Nestin) after receiving either five fractions of 2Gy or a single dose of 10Gy on U87MG and GBM 8401 cells. (**D**) Protein level alterations of CSCs-related genes (CD133, CD44 and Nestin) and autophagic genes (P62 and LC3II) after receiving the indicated radiation doses in 72 h. GAPDH were used as an internal control. Bar graphs represent mean of triplicates ± SD. * *p* < 0.05, ** *p* < 0.01, *** *p* < 0.005 compared with the 2 Gy × 5 times (10 Gy-fraction) group. # *p* < 0.05, ## *p* < 0.01, ### *p* < 0.005 compared with the 10 Gy × 1 time (10 Gy-only) group.

**Figure 4 life-11-00451-f004:**
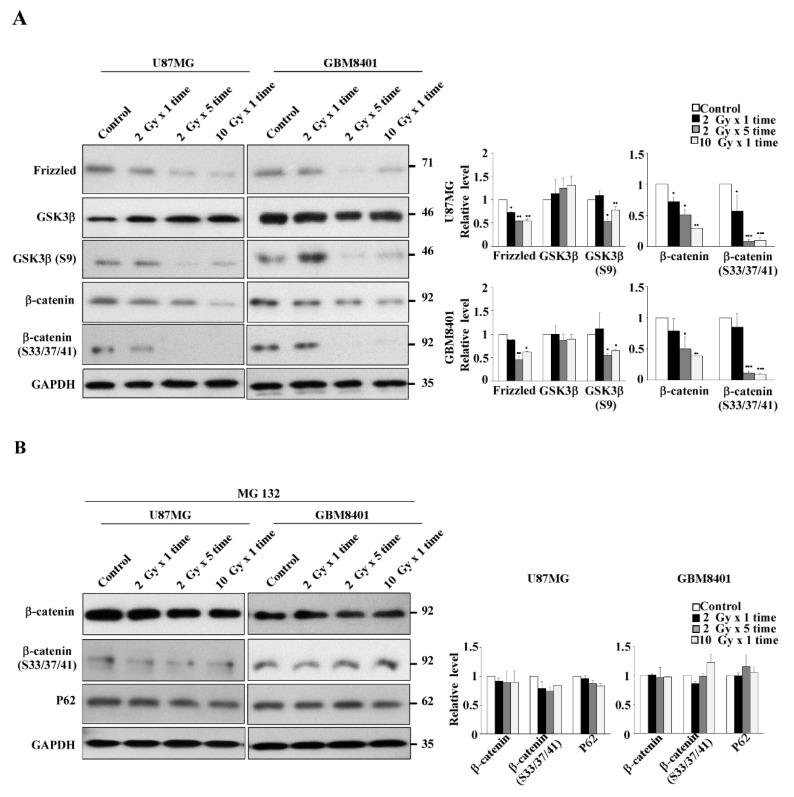
Expression of Wnt/β-catenin/GSK3β pathway under both dose- and fraction-dependent manners in irradiated U87MG and GBM8401 cells. (**A**) GBM8401 cells were treated with radiation for 72 h, and the protein expression patterns of the Wnt pathway were determined. Phospho-β-catenin (Ser33/Ser37/Thr41) was used as β-catenin active forms for detecting the β-catenin status. GSK3β (S9) was used for β-catenin phosphorylation to degrade β-catenin. (**B**) The cells were irradiated with indicated dosages and were then incubated with MG-132 (10 µM) for 8 h. Cytosolic fractions were prepared and subjected to Western blotting with frizzled, GSK3β, phosphor S9 GSK3β, β-catenin, phosphor S33/37/41-β-catenin, and P62 antibodies. GAPDH was used as an internal control. Bar graphs represent the mean of triplicates ± SD. * *p* < 0.05, ** *p* < 0.01, *** *p* < 0.001 compared with the control group.

**Figure 5 life-11-00451-f005:**
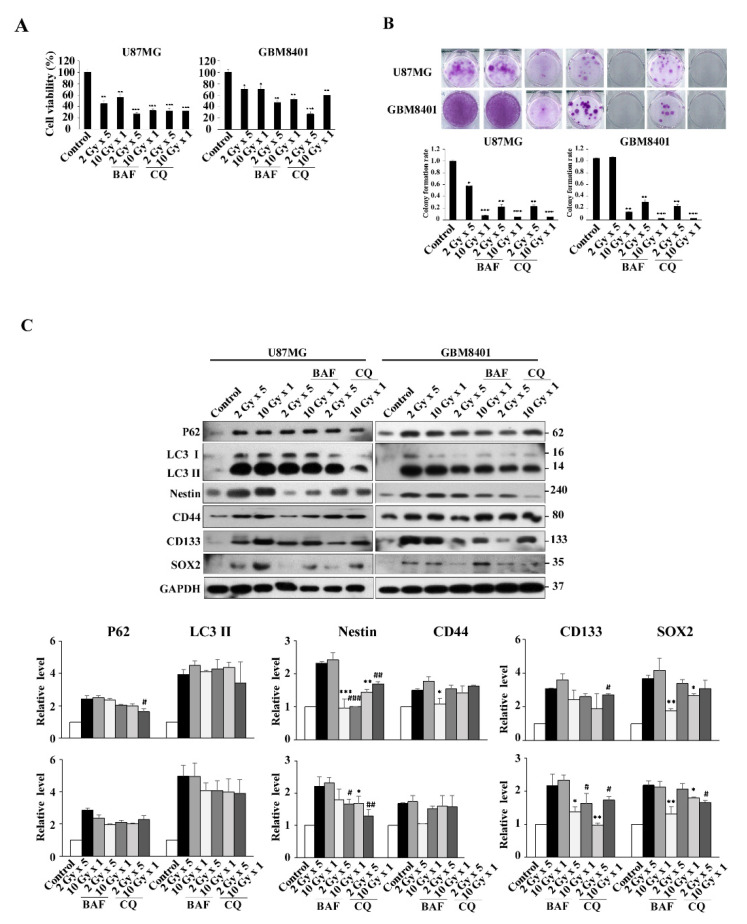
Autophagy inhibition promotes radiation-induced cell death, and reduced radiation induced stemness in U87MG and GBM8401 cells. (**A**) Combination of radiation and autophagy inhibition with BAF (0.1 μM) or CQ (100 μM) decreased the U87MG and GBM8401 cell viability comparing with radiation only. (**B**) Clonogenic assays were performed to assess the effect of irradiation on colony formation in U87MG and GBM8401 cells with or without BAF (0.1 μM) or CQ (100 μM) treatment. The image shows colonies produced by the U87MG and GBM8401 cells following plating of 500 cells and 21 days incubation. Cells were quantified and error bars represent mean ± SEM of three independent experiments. The level of significance was determined using Student’s *t*-Test with ns representing *p* > 0.05, *** *p* < 0.005, ** *p* < 0.01, and * *p* < 0.05 compared with non-irradiated cells without BAF or CQ treatment. (**C**) Transcript levels of autophagic and CSCs-related genes were detected by Western blotting. Cells were irradiated, with or without BAF (0.1 μM) and CQ (100 μM) treatment for 72 h, followed by Western blot analysis of the autophagic and CSCs-related genes. GAPDH was used as an internal control. Bar graph represents mean of triplicates ± SD. * *p* < 0.05, ** *p* < 0.01, *** *p* < 0.005 compared with the 2 Gy × 5 times (10 Gy-fraction) group. # *p* < 0.05, ## *p* < 0.01, ### *p* < 0.005 compared with the single dose 10 Gy (10 Gy-only) group.

**Figure 6 life-11-00451-f006:**
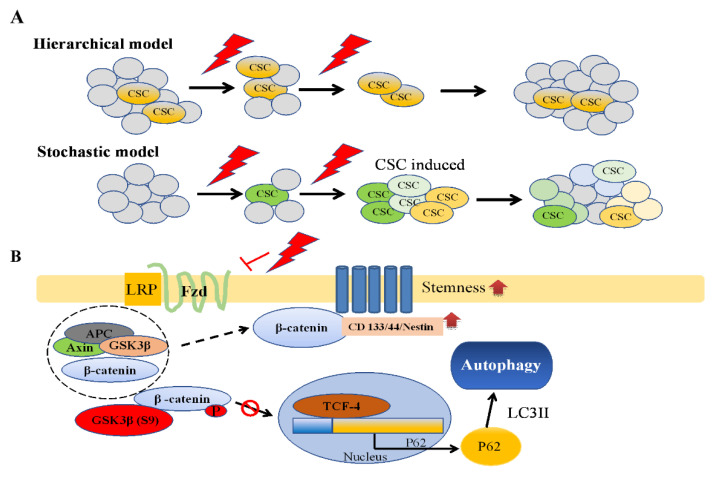
Schematic illustrations for demonstrating the origin of CSCs models and depicting the Fzd/Wnt/β-catenin/P62 axis on ionizing radiation-induced CSCs. (**A**) Origin of CSCs models. Two main models, the hierarchical model and the stochastic model, have been proposed to explain how CSCs evolv. A major point of debate is whether the CSCs exist in the tumor originally or whether conversion between non-CSCs and CSCs could happen. (**B**) Fzd/Wnt/β-catenin/P62 axis working model summarizing the possible pathways for ionizing radiation-induced CSCs makers, such as CD133, CD44, and nestin via P62-mediated autophagy (for details, see Section 4, Discussion).

**Table 1 life-11-00451-t001:** Summary of three cohorts of GBM data sets.

GEO no.	GSE107040	GSE117126	GSE82139
Pubmed ID	29559744	31444412	29088733
Journal	Oncogene	Oncogene	Oncotarget
Overall design	U87MG GBM cells	U-87 MG cells/injected/brain of mice	patient
Brain tumor (GBM)_ non-IR	3	1	6
Brain tumor (GBM)_2Gy	3	1	6
Array platform	Illumina HumanHT-12 V4.0	HuGene-2_0-st	HuGene-1_0-st

**Table 2 life-11-00451-t002:** Correlation between LC3 II expression and clinicopathological parameters in brain tumors.

Clinicopathological Factors	LC3 II				*p*
**Score**	0	1	2	3	
**Age**					0.312
>65	16	19	14	15	
<65	12	12	17	1	
**Gender**					0.221
Male	15	17	18	6	
Female	13	14	13	10	
**Tumor grade**					0.563
I	2	5	3	0	
II	4	8	2	2	
III	5	4	6	2	
IV	17	14	20	12	
**Radiation/chemotherapy**					0.021 *
Yes	8	10	24	15	
No	20	21	7	1	
**KPS ^1^**					0.452
>70	10	15	17	6	
<70	18	16	14	10	

^1^ Karnofsky Performance scale (KPS). * *p* < 0.05, statistically significant difference. The data analyzed using one ANOVA.

## Data Availability

Data is contained within the article or Appendix A.

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
