# Peer review of "Ionizing Radiation Induces Resistant Glioblastoma Stem-Like Cells by Promoting Autophagy via the Wnt/β-Catenin Pathway"

_life, 2021, doi:10.3390/life11050451_

Round 1

Reviewer 1 Report

In the manuscript, the authors used human GBM samples and they have not mentioned anything related to ethical approval in the methods sections.

Before considering for a review the authors should revise the manuscript with proper ethical committee approvals (Informed consent for research)

Author Response

Responses to reviewer comments:

Response to reviewer 1:

To answer point-by-point the details of the revisions to reviewer 1:

  1. In the manuscript, the authors used human GBM samples and they have not mentioned anything related to ethical approval in the methods sections.

Answer 1: Sorry for our mistake. We already added the IRB number for ethical approval in lines 103-106.

  1. Before considering for a review the authors should revise the manuscript with proper ethical committee approvals (Informed consent for research).

Answer 2: We already revised the manuscript for proper ethical committee approvals as “The protocols and ethics statements in this study were approved by the Institutional Review Board of Kaohsiung Medical University Hospital (No. KMUHIRB-20110185 and No. KMUHIRB-F(I)-20200024). Informed consent was obtained from all subjects involved in the study in lines 103-106.

Reviewer 2 Report

The manuscript of Yu and co-workers in interesting paper on irradiation-resistant mechanisms of glioblastoma cells. However, the English needs extensive editing as parts of the manuscript are written in a manner that the text is not clear and understandable in a way to understand the experiments and conclusions correctly. Moreover, there are several issues regarding methodology and presentation of results. Based on current text, the conclusions are not supported by the methodology (also due to the fact that methodology and result sections are no written in a clear way).

Other comments:

-Images are of poor quality and are hard to interpret.

-Ethical part is not written, do you have institutional or national ethical approvals for collecting patient data and tissues?

-How was the scoring of IHC performed on tissue slides? Not written, what does 0-3 scores mean?

-qPCR LC3II evaluation, which gene of official nomenclature do you mean? MAP1LC3A/B or something else?

-How was quantification of Western blot performed?

-By dewaxing, you mean deparaffinization of slides? 

-Database analyses: used only 1-2 cases of xenografts and patient tissue, low number of samples to make conclusions

-Clinicopathological data correlation: poor description of glioma patients, usually all glioma patients are treated at least with chemo or radiotherapy. Also, it is not clear what does radiation/chemotherapy means? Were these patients treated with both or just irradiation or just chemo? Whuch chemotherapy was performed?

-Comparison of IHC staining in pre and post treated samples: were these samples obtained from the same patients, not clear. If not, the differential expression can be a consequence of inter-patient variability.

-Can you quantify IHC staining of stem markers and autophagy markers? It is impossible to make conclusion based on 1 image.

-Results are poorly explained in some segments, including page 8, last paragraph, page 10 first paragraph.

-Figure 4B, I cannot see results in control samples, without MG132.

-Statistical analyses, which program or software was used?

Author Response

Responses to reviewer comments:

Response to reviewer 2:

The manuscript of Yu and co-workers in interesting paper on irradiation-resistant mechanisms of glioblastoma cells. However, the English needs extensive editing as parts of the manuscript are written in a manner that the text is not clear and understandable in a way to understand the experiments and conclusions correctly. Moreover, there are several issues regarding methodology and presentation of results. Based on current text, the conclusions are not supported by the methodology (also due to the fact that methodology and result sections are no written in a clear way).

To answer point-by-point the details of the revisions to reviewer 2:

  1. -Images are of poor quality and are hard to interpret.

Answer 1: We already offered the high quality of images (tiff) in first submission. You could ask for the original images from editor office.

  1. -Ethical part is not written, do you have institutional or national ethical approvals for collecting patient data and tissues?

Answer 2: We already revised the manuscript for proper ethical committee approvals as

“The protocols and ethics statements in this study were approved by the Institutional Review Board of Kaohsiung Medical University Hospital (No. KMUHIRB-20110185 and No. KMUHIRB-F(I)-20200024). Informed consent was obtained from all subjects involved in the study in line 103-106.

  1. -How was the scoring of IHC performed on tissue slides? Not written, what does 0-3 scores mean?

Answer 3: LC3II expression was grading by scoring system (0 is no expression, 1 is mild expression, 2 is moderate expression and 3 is strong expression).

  1. -qPCR LC3II evaluation, which gene of official nomenclature do you mean? MAP1LC3A/B or something else?

Answer 4: The official nomenclature of our LC3II in qPCR LC3II evaluation is Homo sapiens microtubule associated protein 1 light chain 3 beta 2 (MAP1LC3B2), and the NCBI reference sequence is NM_001085481.

  1. -How was quantification of Western blot performed?

Answer 5: We corrected it as “The bands were quantified using the Multi Gauge 3.0 program (Fujifilm, Tokyo, Japan). In brief, we use the rectangle tool to select indicated region of protein signal which the software automatically reports the intensity of signal. Then divide the protein signal intensity by internal control intensity to obtain the relative ratio.” In lines 165-169.

  1. -By dewaxing, you mean deparaffinization of slides? 

Answer 6: Yes. We used deparaffinization of slides.

  1. -Database analyses: used only 1-2 cases of xenografts and patient tissue, low number of samples to make conclusions

        Answer 7: Your question is well taken. We used not only three data bases to prove out hypothesis but also used IHC data from 2 GBM patients. These data could be enough for our initial hypothesis. Then we performed experiments to make the conclusion. 

  1. -Clinicopathological data correlation: poor description of glioma patients, usually all glioma patients are treated at least with chemo or radiotherapy. Also, it is not clear what does radiation/chemotherapy means? Were these patients treated with both or just irradiation or just chemo? Which chemotherapy was performed?

Answer 8: Sorry for our mistake. We corrected it as “All patients were also separate into two groups as radiation or chemotherapy and without any therapy. Temozolomide (TMZ) was used as chemotherapy. Retrospective analysis showed that GBM patients who received radiation or chemotherapy had significantly higher levels of LC3II expression, when compared with those without receiving radiation or chemotherapy (P = .021, Table 2), meaning autophagic markers is related to radiation/chemotherapy.” In lines 109-113.

  1. -Comparison of IHC staining in pre and post treated samples: were these samples obtained from the same patients, not clear. If not, the differential expression can be a consequence of inter-patient variability.

Answer 9: Yes, IHC staining in pre and post treated samples is from the same patient. So inter-patient variability could be excluded.

  1. -Can you quantify IHC staining of stem markers and autophagy markers? It is impossible to make conclusion based on 1 image.

Answer 10: Your question is well taken. Due to the fewer cases of GBM, we could only provide the representative case to make the conclusion. Therefore, we used three cohort data bases and clinical GBM patient’s data prove our initial hypothesis. Then we performed GBM cell lines for further experiments to make the conclusion.

  1. -Results are poorly explained in some segments, including page 8, last paragraph, page 10 first paragraph.

Answer 11: Clongenic assay revealed regrowth after IR treatment. It implies the regrowth cells as inductive IR-resistant cell (page 8, last paragraph). Then we further demonstrated the inductive IR-resistant cell presented as cancer stem cell markers (page 10 first paragraph).

  1. -Figure 4B, I cannot see results in control samples, without MG132.

Answer 12: Figure 4A is for control groups, without MG132. Figure 4A is for the effect of IR and figure 4B is based on Figure 4A. We used MG132 as proteasome inhibitor to check if the amount of beta-catenin phosphor-form (33/37/41) was reversed. We thought it is enough to solve the problem of internal control. Figure 4A is internal control for Figure 4B.

  1. -Statistical analyses, which program or software was used?

Answer 13: We corrected it as “The following statistical analysis is calculated on SPSS (Version 22). For the in-vitro and in-vivo studies, statistical significance was evaluated through one-way analysis of variance (ANOVA) followed by Tukey’s post hoc test to correct for multiple comparisons. A two-tailed Student’s t-test was used to compare data between two groups. Statistical significance was set as p < 0.05. The association between P62, LC3 II and CD133 expressions and clinicopathological characteristics of GBM patients were investigated using ANOVA test, Fisher's exact test, as well as Chi-squared test. Survival analysis was performed using the Kaplan–Meier method to calculate overall and dis-ease-free survival rates among different groups. These groups were then compared using log-rank method. Two-sided p < 0.05 was considered as statistically significant difference.” In lines 197-206.

Reviewer 3 Report

Tsai et al report a study where they look at expression of glioma stem cell markers and induction of autophagy in GBM human tissues and in GBM cell lines. Based on the observations from IHC of human samples author investigate whether irradiation induces stemness in GBM and whether this coincides with induction of autophagy. Authors then combined these investigation with using inhibitors of autophagy, bafilomycin (BAF) and chloroquine (CQ).

Overall, this work is of some interest to the field, confirming that radiotherapy is inducing stemness in GBM and also that this might be through induction of autophagy. However there are some issues with this manuscript and work. First of all, the language is very poor and makes it difficult to read. This manuscript requires language corrections from a professional English speaker. Even the abstract is full of incorrect phrases and somehow wrong statements, which are huge shortcuts in what authors really were trying to say. The manuscript have been so carelessly submitted, even with the notes that authors wrote to themselves in the process of writing the manuscript, e.g. in line 67 page 2 of the pdf, authors say: „… to cytotoxic agents, including IR and TMZ (spell out abbreviation), in GBM treatment”. Moreover more specific points to various parts of the manuscript are listed below.

In the methods section I could not find any information about the ethical committee approval for the use of 106 human brain tissues from “patients with stage I–IV glioma in Department of Neurosurgery, Kaohsiung Medical University Hospital between 2000-2011”.

Figure 1A, what is on the  “X” axis? Days?

What authors mean by saying: “expression of P62 and LC3II proteins were lower in U87MG and H4 cell lines than in their parental cell lines (U251, T98G, GBM8401 and MO59K; Supplement Figure 2).” in the lines 266-268? For example, how U251 cell line is parental to U87MG?

In Figure 3B – how time-point „0 h” was performed? Cells were Irradiated and harvested immediately? Is this a qPCR? For the Figure 4D – how the quantification of densitometry was performed for LC3II? Normally it is shown as a ratio of LC3II/LC3I forms normalized to the loading control, which can represent the rate of autophagy. In general in figure legends, it is not clear how the quantifications of blots were calculated and normalized for all other antibodies.

GAPDH blots are substantially oversaturated, particularly in Figure 4, which makes it difficult to judge the actual changes between the samples tested.

Experiment in Figure 4B is missing a no-MG-132 control, and so it is difficult to judge if changes/lack of changes in beta-Catenin are due to different saturation of bands between the figures 4A and 4B.

The pictures of the clonogenic experiment (Fig. 2B) suggest that colonies are overgrown in some wells and hence disqualify these wells for the quantification. Similarly in Figure 5, some wells are just purple and no single colonies are seen that could be quantified.

It would be interesting to see the effect of BAF or CQ only on the survival of GBM cells and expression of GSC markers without adding IR doses (Figure 5)? This control is missing.

Authors discuss hierarchical and stochastic models of the origin of cancer stem cells after IR. However, based on the data presented it is difficult to rule out any of the models and authors imply that their data suggests more stochastic model of the origin of CSCs in GBM. I think that this conclusion is overstated.  

Author Response

Responses to reviewer comments:

Response to reviewer 3:

Tsai et al report a study where they look at expression of glioma stem cell markers and induction of autophagy in GBM human tissues and in GBM cell lines. Based on the observations from IHC of human samples author investigate whether irradiation induces stemness in GBM and whether this coincides with induction of autophagy. Authors then combined these investigation with using inhibitors of autophagy, bafilomycin (BAF) and chloroquine (CQ).

To answer point-by-point the details of the revisions to reviewer 3:

  1. Overall, this work is of some interest to the field, confirming that radiotherapy is inducing stemness in GBM and also that this might be through induction of autophagy. However, there are some issues with this manuscript and work. First of all, the language is very poor and makes it difficult to read. This manuscript requires language corrections from a professional English speaker. Even the abstract is full of incorrect phrases and somehow wrong statements, which are huge shortcuts in what authors really were trying to say. The manuscript has been so carelessly submitted, even with the notes that authors wrote to themselves in the process of writing the manuscript, e.g. in line 67 page 2 of the pdf, authors say: „… to cytotoxic agents, including IR and TMZ (spell out abbreviation), in GBM treatment”. Moreover, more specific points to various parts of the manuscript are listed below.

Answer 1:

  1. A) Your comment is well taken. We already made fully English edition for English language by Wallace Academic Editing.
  2. B) We also modified our abstract with correct phrases in lines 24-29.
  3. C) We also revised the “in line 67 page 2 of the pdf, authors say: „… to cytotoxic agents, including IR and TMZ (spell out abbreviation)” to correct form as “Temozolomide (TMZ)” in line 68
  4. In the methods section I could not find any information about the ethical committee approval for the use of 106 human brain tissues from “patients with stage I–IV glioma in Department of Neurosurgery, Kaohsiung Medical University Hospital between 2000-2011”.

Answer 2: We already revised the manuscript for proper ethical committee approvals as “The protocols and ethics statements in this study were approved by the Institutional Review Board of Kaohsiung Medical University Hospital (No. KMUHIRB-20110185 and No. KMUHIRB-F(I)-20200024). Informed consent was obtained from all subjects involved in the study” in line 103-106.

  1. Figure 1A, what is on the “X” axis? Days?

Answer 3: X axis means “Weeks”. We also corrected it in Figure 1A.

  1. What authors mean by saying: “expression of P62 and LC3II proteins were lower in U87MG and H4 cell lines than in their parental cell lines (U251, T98G, GBM8401 and MO59K; Supplement Figure 2).” in the lines 266-268? For example, how U251 cell line is parental to U87MG?

    Answer 4: Sorry for our mistake, we changed it as “other GBM cell lines” in line 270.

  1. In Figure 3B – how time-point „0 h” was performed? Cells were Irradiated and harvested immediately? Is this a qPCR? For the Figure 4D – how the quantification of densitometry was performed for LC3II? Normally it is shown as a ratio of LC3II/LC3I forms normalized to the loading control, which can represent the rate of autophagy. In general in figure legends, it is not clear how the quantifications of blots were calculated and normalized for all other antibodies.

Answer 5:

  1. A) 0 h is for irradiated and harvested immediately after IR treatment.
  2. B) in Figure 3B. It is qPCR data.
  3. C) According to Autophagy Review. How to Interpret LC3 Immunoblotting. Pages 542-545 | Published online: 29 Jun 2007- Issue 6 and Guidelines for the use and interpretation of assays for monitoring autophagy (4th 2021). LC3II is the main index for autophagic marker, but not LC3I. Therefore, we thought our original LC3II data is more suitable in our revision manuscript.

However, we also made the LC3II/LC3I ratio which is calculated as you requested. Comparing with our original LC3II data, the result showed the same conclusion.

  1. D) Anti-GAPDH or anti-β-actin was used in all lanes as internal loading control and also as a fixed value of 1 to which all calculations for each lane were standardized. The bands were quantified using the Multi Gauge 3.0 program (Fujifilm, Tokyo, Japan).
  2. GAPDH blots are substantially oversaturated, particularly in Figure 4, which makes it difficult to judge the actual changes between the samples tested.

Answer 6: We choose another blot with less saturated internal control (Upper panel, new) to replace oversaturated blot (lower panel, old).

  1. Experiment in Figure 4B is missing a no-MG-132 control, and so it is difficult to judge if changes/lack of changes in beta-Catenin are due to different saturation of bands between the figures 4A and 4B.

Answer 7: The protein concentration, primary antibody dilution and treating condition between figure 4A and B are the same. Please also check the answer 6, we provided the more suitable data for GAPDH. Figure 4A is for the effect of IR and figure 4B is based on Figure 4A. We used MG132 as proteasome inhibitor to check if the amount of beta-catenin phosphor-form (33/37/41) was reversed. We thought it is enough to solve the problem of internal control. Figure 4A is internal control for Figure 4B

  1. The pictures of the clonogenic experiment (Fig. 2B) suggest that colonies are overgrown in some wells and hence disqualify these wells for the quantification. Similarly in Figure 5, some wells are just purple and no single colonies are seen that could be quantified.

Answer 8: We agree with your comment. The difference of each clonogenic assay is relative pattern. So, if 21-days is too overgrown, 7-days clonogenic assay could be in consideration

  1. It would be interesting to see the effect of BAF or CQ only on the survival of GBM cells and expression of GSC markers without adding IR doses (Figure 5)? This control is missing.

Answer 9: Thanks for your interesting idea and we will take it into consideration. GBM cells will survive and express GSC markers after IR doses. While BAF or CQ was used, GBM cells will not survive and express GSC markers. Therefore, we think it could be no difference in autophagy markers by using BAF or CQ without IR doses.

  1. Authors discuss hierarchical and stochastic models of the origin of cancer stem cells after IR. However, based on the data presented it is difficult to rule out any of the models and authors imply that their data suggests more stochastic model of the origin of CSCs in GBM. I think that this conclusion is overstated. 

Answer 10: We agree with your comment. We know the origin of cancer stem cell is challenging issue and difficult to solve. Therefore, we changed our conclusion as “Although our data revealed that CD133/Cd44/Nestin/SOX2 was increased by IR, whether stochastic model or hierarchical model is more suitable for GSC by IR still need evidences to prove.” In lines 434-436.

Round 2

Reviewer 2 Report

The authors improved the manuscript according to suggestions, however there are still some issues that need to be addressed before acceptance.

-It is still not clear how LC3III expression was quantified, what does mild, moderate and strong expression means? Are they based on intensity or frequency of staining (% of cells positive)? Please elaborate.

-How many GBM samples did you use for IHC analysis of stem cell and autophagy markers before and after treatment? Please add this information (Figure 1).

-It seems to me that LC3III IHC staining is higher also in pretreated GBM samples. Please correct the result section (Figure 1). Moreover, it looks that some IHC staining are non-specific in Figure 1 (LC3III and CD133 after treatment). Did you perform control stainings? Which? To confirm non-specific IHC staining of GBM tissues.

-Official gene names should be included in Methods section (qPCR).

Author Response

Responses to reviewer comments:

Response to reviewer 2:

To answer point-by-point the details of the revisions to reviewer 2:

  1. It is still not clear how LC3III expression was quantified, what does mild, moderate and strong expression means? Are they based on intensity or frequency of staining (% of cells positive)? Please elaborate.

Answer 1:

Thanks for your comment. LC3II expression is based on the following criteria:

Frequency of nucleus staining <10% of cell positive                   

score is 0

Frequency of nucleus staining > 10% of cell positive and intensity is low        

score is 1

Frequency of nucleus staining > 10% of cell positive and intensity is moderate    

score is 2

Frequency of nucleus staining >10% of cell positive and intensity is strong      

score is 3

We also corrected it as “0 is frequency of nucleus staining <10% of cell positive; 1 is frequency of nucleus staining > 10% of cell positive and intensity is low; 2 is frequency of nucleus staining > 10% of cell positive and intensity is moderate; 3 is frequency of nucleus staining >10% of cell positive and intensity is strong” in lines 109-112.

  1. How many GBM samples did you use for IHC analysis of stem cell and autophagy markers before and after treatment? Please add this information (Figure 1).

Answer 2: Your question is well taken. We used 2 representative GBM patients for IHC analysis of stem cell and autophagy markers before and after treatment. We also added this information in Figure 1 (in line 254 and 256) and result 3.1 (in line 238-239).

  1. It seems to me that LC3III IHC staining is higher also in pretreated GBM samples. Please correct the result section (Figure 1). Moreover, it looks that some IHC staining are non-specific in Figure 1 (LC3III and CD133 after treatment). Did you perform control stainings? Which? To confirm non-specific IHC staining of GBM tissues.

Answer 3: Thanks for your comment.

  1. a) Although LC3II staining for the cytoplasm (back-ground staining) in pretreated GBM samples seems equal to post-treated GBM samples, we used frequency of nucleus staining for final result. The frequency of nucleus staining for LC3II in post-treat GBM sample is more than pretreated samples. Therefore, we though LC3II IHC staining is still increased in post-treated GBM samples (Reliable LC3 and p62 Autophagy Marker Detection in Formalin Fixed Paraffin Embedded Human Tissue by Immunohistochemistry. Eur J Histochem. 2015 Apr 13; 59(2): 2481).
  2. b) We used hematoxylin (H&E) and Glial fibrillary acidic protein (GFAP) for control staining to confirm non-specific IHC staining of GBM tissues. For LC3II group after treatment, we focus on nucleus staining. For CD133 group after treatment, we focus on cytoplasmic and membranous staining. Therefore, each group has each specific target.

Immunohistochemical studies:

   IDH1 R132H: negative.

   ATRX: expression loss in 80% of tumor cells.

   p53: diffuse positive.

   Ki-67: 60 %.

   MGMT: expression loss in 70% of tumor cells.

   GFAP: positive.

   CK: negative.

  1. Official gene names should be included in Methods section (qPCR).

Answer 4: Thanks for your comment. We added the sentence as “The primer sequences were as follows: CD133 (Homo sapiens pan paniscus prominin 1, PROM1/CD133, NCBI reference sequence is XM_034958337.1) forward 5'-GAGCTAAGGGAAGGGCGG-3'; CD133 reverse 5'-TTCTGTCTGAGGCTGGCTTG-3'; CD44 (Homo sapiens pan paniscus CD44, NCBI reference sequence is XM_034932399.1) forward 5'-CTGCAGGTATGGGTTCATAG-3'; CD44 reverse 5'-ATATGTGTCATACTGGGAGGTG-3'; Nestin (Homo sapiens trachypithecus francoisi nestin, NES, NCBI reference sequence is XM_033196923 ) forward 5'-GGGAAGAGGTGATGGAACCA-3'; Nestin reverse 5'-AAGCCCTGAACCCTCTTTGC-3'; P62 (Homo sapiens sapajus apella sequestosome 1, SQSTM1/P62, NCBI reference sequence is XM_032270231.1) forward 5'-TACCAGGACAGCGAGAGGAA-3'; P62 reverse 5'-TCCTTTCTCAAGCCCCATGT -3'; LC3II (Homo sapiens microtubule associated protein 1 light chain 3 beta 2, MAP1LC3B2, NCBI reference sequence is NM_001085481) forward 5'-GATGTCCGACTTATTCGAGAGC-3'; LC3II reverse 5'-TTGAGCTGTAAGCGCCTTCTA-3'; 18s (Homo sapiens DIMT1 rRNA methyltransferase and ribosome maturation factor, DIMT1/18s,NCBI reference sequence is NM_001348076.2) forward 5'-TCAAGTGCAGTGCAACAACTC-3'; 18s reverse 5'-AGAGGACAGGGTGGAGTAATCA-3'.” in lines 144-159.

Reviewer 3 Report

Authors addressed most of my comments. However, it still is not clear how authors quantified saturated wells in the clonogenics assay.

Author Response

Responses to reviewer comments:

Response to reviewer 3:

To answer point-by-point the details of the revisions to reviewer 3:

  1. Authors addressed most of my comments. However, it still is not clear how authors quantified saturated wells in the clonogenics assay.

Answer 1: Thanks for your comment. We added the sentence as “Then the cells were washed with PBS, then DMSO was added to induce a completely dissolution of the crystal violet. Absorbance was recorded at 570 nm by a 96-well-plate ELISA reader.” in lines 128-130.
